# Effects of Dietary Multienzyme Complex Supplementation on Growth Performance, Digestive Capacity, Histomorphology, Blood Metabolites and Hepatic Glycometabolism in Snakehead (*Channa argus*)

**DOI:** 10.3390/ani12030380

**Published:** 2022-02-04

**Authors:** Xiaoqing Ding, Xinzheng Nie, Chunchun Yuan, Lai Jiang, Wenxin Ye, Lichun Qian

**Affiliations:** 1Key Laboratory of Animal Nutrition and Feed Science in East China, Ministry of Agriculture, College of Animal Sciences, Zhejiang University, Hangzhou 310058, China; 21817065@zju.edu.cn (X.D.); 22017006@zju.edu.cn (C.Y.); 2Shandong Animal Husbandry General Station, Jinan 250022, China; 15615588008@163.com; 3Hainan Academy of Zhejiang University, Zhejiang University, Sanya 572025, China; 22017078@zju.edu.cn (L.J.); 22117080@zju.edu.cn (W.Y.)

**Keywords:** multienzyme complex, growth performance, gastrointestinal function, glycometabolism, metabolomics, snakehead (*Channa argus*)

## Abstract

**Simple Summary:**

The multienzyme complex is composed of one or several single enzymes as the main component, mixed with other single enzyme preparations, or obtained by the fermentation of one or more microorganisms, and an the exogenous enzyme mixture with multiple functions of a single enzyme. Supplementation with exogenous enzyme preparations offsets the negative effects of removing antibiotics from animal diets. The multienzyme complex supplemented in aquatic feed can enhance the secretion and activity of endogenous digestive enzymes in aquatic animals. Meanwhile, exogenous digestive enzymes can also help aquatic animals to decompose some anti-nutrients, improve the utilization and digestibility of feed, and promote animal growth.

**Abstract:**

The present study evaluated the impact of dietary multienzyme complex (MEC) supplementation on growth performance, digestive enzyme activity, histomorphology, serum metabolism and hepatopancreas glycometabolism in snakeheads (*Channa argus*). A total of 600 fish (initial weight, 69.70 ± 0.30 g) were randomly divided into four groups. Four diets were formulated: (1) control (basic diet); (2) E1 (400 U kg^−1^ amylase, 150 U kg^−1^ acid protease, 1900 U kg^−1^ neutral protease and basic diet); (3) E2 (800 U kg^−1^ amylase, 300 U kg^−1^ acid protease, 3800 U kg^−1^ neutral protease and basic diet); and (4) E3 (1200 U kg^−1^ amylase, 450 U kg^−1^ acid protease, 5700 U kg^−1^ neutral protease and basic diet). The results show that the E2 group increased the specific growth rate, weight gain rate and the final body weight, as well as decreasing the blood urea nitrogen, alanine aminotransferase and triglyceride. The mRNA levels and activities of digestive enzymes and key glucose metabolism enzymes in the hepatopancreas were enhanced in snakeheads fed the MEC. Meanwhile, moderate MEC diet (E2 groups) supplementation improved digestive tract morphology, increased the glycogen in the hepatopancreas and the lipids in the dorsal muscle. Moreover, plasma metabolomics revealed differential metabolites mainly involved in amino acid metabolism. These findings suggest that dietary supplementation with the MEC improved growth performance, digestive tract morphology, gene expression and the activity of digestive enzymes, enhanced the glycolysis-gluconeogenesis and amino acid metabolism of snakeheads, and the optimal composition of the MEC was group E2.

## 1. Introduction

The snakehead (*Channa argus*), as an important invasive species, is mainly distributed in Africa and Asia, and has been regarded as a traditional high-quality edible fish in China due to its taste, high nutritional value, and its recuperative and medicinal qualities [1,2]. However, the cultivation of snakeheads is highly dependent on feeding on trash fish and a diet consisting of animal raw materials, and snakeheads have a shorter digestive tract and fewer insulin receptors than other fishes, resulting in slower carbohydrate breakdown and incomplete feed absorption [2,3]. Recently, with the high demand and rising price of trash fish as well as ecological and environmental issues, it is of great significance to develop suitable feed additives to maximize the nutritional value of the feed and to support the sustainable supply of snakeheads.

The multienzyme complex (MEC) is composed of one or several single enzymes as the main component, mixed with other single enzyme preparations, or obtained by the fermentation of one or more microorganisms [4]. It has been reported that supplementation with exogenous enzyme preparations offsets the negative effects of removing antibiotics from animal diets [5]. The MEC supplemented into aquatic feed can enhance the secretion and activity of endogenous digestive enzymes in aquatic animals [6]. Meanwhile, exogenous digestive enzymes can also help aquatic animals to decompose some anti-nutrients, improve the digestibility and utilization of feed, and promote animal growth [7,8]. Our previous studies have shown that snakeheads fed a combination with conventional feeds may have damaged and deficient villi due to incomplete feed absorption [9]. However, the supplementation of amylase to snakehead feed was thought to promote starch utilization and alleviate barriers to carbohydrate consumption [10]. In addition, acid proteases exhibited reliable activity under acidic conditions, and supplementation of acid proteases into feed could facilitate absorption and digestion of chyme in the stomach [11].

Although exogenous enzymes have been widely applied in the poultry and livestock feed industries as feed additives [12,13], studies on exogenous enzyme supplements for aquaculture species has been focused on phytase [14]. There is little research and information on the effects of the MEC on the hepatic material metabolism, histological morphology and gene expression of aquatic animals, especially snakeheads. Thus, we hypothesized that supplementation of the MEC can improve growth performance, digestive enzyme activity, hepatic metabolism and histological morphology of the snakehead. Additionally, metabolomics is an emerging research field that uses high-throughput technology to quantify small-molecule substrates, intermediates, and metabolites in biological samples, and elucidate metabolic changes and nutritional intervention mechanisms in biological systems through integration with related metabolites [15,16]. Here, we carried out metabolomics analysis, and preliminarily studied the changes in the snakehead serum metabolites after feeding with MEC feed to identify the supplementary effects of the MEC.

## 2. Materials and Methods

### 2.1. Preparation of Diets

The enzymes used in this experiment was obtained from Huaqi Biotechnology Co., Ltd. (Guangzhou, China), and the enzyme activity of amylase, acid protease and neutral protease were 6700, 8000, and 50,000 U kg^−1^, respectively. The approximate composition of the diet was measured according to the AOAC [17] standard method. All diets contained 49.5 g/kg moisture, 66.5 g/kg ash, 17.6 g/kg crude fiber, 135.0 g/kg ether extract, and 431.7 g/kg crude protein. In addition to the composition of the four diets and the total energy (21.53 MJ/kg) measured with an oxygen bomb calorimeter (Parr Instrument Company, Moline, Illinois, USA), the results of the compositional analysis of the experimental diets are shown as percentages on a dry matter basis in Table 1. The optimal combination of amylase, acid protease and neutral protease was designed using the Box–Behnken Design of Response Surface and determined by in vitro digestion. We finally confirmed that the optimization enzymes for amylase, acid protease and neutral protease were 800, 300 and 3800 U kg^−1^, respectively. The diets were processed into pellets with a diameter of 5 mm using a steam extruder (DGP80-II; Yugong Technology Development Co., Ltd., Hebei, China). The MEC was sprayed on the diet after puffing, and the outlet temperature was 80 ± 5 °C.

### 2.2. Animals and Sampling Procedures

After acclimatization, a total of 600 fish (initial weight, 69.70 ± 0.30 g) were selected, and the fish were randomly divided into 4 groups, which included 5 net-cages (100 cm × 100 cm × 180 cm) for each treatment—the cage was 20 cm above the water surface, and each net-cage contained 30 fish. Four diets were formulated: (1) control group, fed with basic diet; (2) E1 group, fed with enzymes for 400 U kg^−1^ amylase, 150 U kg^−1^ acid protease, 1900 U kg^−1^ neutral protease and basic diet; (3) E2 group, fed with enzymes for 800 U kg^−1^ amylase, 300 U kg^−1^ acid protease, 3800 U kg^−1^ neutral protease and basic diet; (4) E3 group, fed with enzymes for 1200 U kg^−1^ amylase, 450 U kg^−1^ acid protease, 5700 U kg^−1^ neutral protease and basic diet. All the fish were fed at 3–5% of their body weight per day, and were fed twice a day at 8:00 and 16:00 h for 60 d. Feeding was performed until apparent satiety, and we adjusted the feeding level over time according to environmental conditions such as fish growth, feeding situation and water temperature. After the breeding experiment, the fish were fasted for 24 h, and 8 fish were randomly selected from each cage. The fish were anesthetized with MS-222 solution (120 mg/L) and blood was collected from the tail vein. Parts of the stomach, intestines, organs and hepatopancreas were fixed with 4% paraformaldehyde and 2.5% glutaraldehyde, respectively, for morphological analysis. The other organs and carcasses (no internal organs) were quickly collected from each individual fish on an ice-cold surface, divided into equal portions and quickly frozen with liquid nitrogen. Serum samples and frozen tissues were preserved at −80 °C for further analysis.

### 2.3. Analysis of Serum Biochemistry Index

The levels of glucose (GLU), albumin (ALB), total protein (TP), aspartate aminotransferase (AST), alanine aminotransferase (ALT), blood urea nitrogen (BUN), total cholesterol (TC), and triglyceride (TG) were determined using the glucose kit (A154-1-1; glucose oxidase method), the total protein assay kit (A045-4-2; with standard: BCA method), the albumin assay kit (A028-2-1), the urea assay kit (C013-2-1), the alanine aminotransferase assay kit (C009-2-1), the aspartate aminotransferase assay kit (C010-2-1), the triglyceride assay kit (A110-1-1), and the total cholesterol assay kit (A111-1-1), respectively. All kits were purchased from Jiancheng Bioengineering Institute Co., Ltd. (Nanjing, China).

### 2.4. Glycometabolism and Digestive Enzymes Analysis

The activities of phosphoenolpyruvate carboxykinase (PEPCK), pyruvate kinase (PK), phosphofructokinase (PFK), hexokinase (HK), pyruvate carboxylase (PC), glucose 6-phosphatase (G6P), fructose 1,6-diphosphatase (FBP), glucose phosphate dehydrogenase (G6PDH) and glycogen synthase (GCS) in the hepatopancreas were determined using the relative assay kits (Suzhou Comin Biotechnology Co., Ltd. Suzhou, China). The activities of amylase, pepsin, trypsin, content of hepatopancreas glycogen as well as protein concentration were analyzed using the methods described in the detection kits (Nanjing Jiancheng Bioengineering Institute, Nanjing, China).

### 2.5. Quantitative Real-Time PCR Analysis

Primers were designed using Primer 5.0 software, and the specific primer sequences are shown in Appendix A. Applied Biosystems 7500 PCR (ThermoFisher, Scientific, Carlsbad, NM, USA) was used to determine mRNA relative expression. The PCR reaction system consisted of: cDNA 2 μL, upstream and downstream primers 0.3 μL, 2 × SYBRGreen 10 μL, ddH_2_O 7.4 μL, the total volume of PCR reaction 20 μL. Reaction conditions: pre-denaturation at 95 °C for 5 min, denaturation at 95 °C for 30 s, annealing at 57 °C for 30 s, extension at 72 °C for 30 s, 40 cycles. Quantitative results were calculated and analyzed following the 2^−ΔΔCt^ method as previously described [18].

### 2.6. Analysis of Hepatopancreas Morphology and Dorsal Muscle Component

#### 2.6.1. Hematoxylin and Eosin (H&E) Staining

First, the intestinal and hepatopancreas tissues were dehydrated using the ethanol gradient method (75%, 85%, 95%, 100%), xylene twice for 10 min and paraffin wax three times for 60 min. The tissue in paraffin was embedded with an embedding machine. Then, the tissues sections of 4 μm thickness were sliced in a slicer, flattened on glass slides and baked in a 60 °C oven. Hematoxylin staining was performed for 15 min, and eosin staining was performed for 1 min, followed by dehydration and mounting. Images were acquired using a DM3000 microscope (Leica, Wetzlar, Germany). The villus height was measured using Image-Pro software (Media Cybernetics, Bethesda, MD, USA).

#### 2.6.2. Oil Red O Staining

The OCT-embedded hepatopancreatic tissue samples were cut into 5 μm-thick sections with a cryostat. The sections were stained in pre-warmed Oil Red O staining solution at 60 °C for 10 min. After isopropanol differentiation, the cells were washed with distilled water and counterstained with hematoxylin. Stained sections were visualized and imaged with a DM3000 microscope (Leica, Wetzlar, Germany). The chemical composition of the dorsal muscle, including crude protein, crude lipid and crude ash, were analyzed following a standard method [17].

### 2.7. Electron Microscopy Examinations

For scanning electron microscopy, the glutaraldehyde-fixed tissue was left for one night and then fixed with 1% OsO_4_ for 1 h. The intestinal samples were washed three times with 0.01 M PBS buffer and dehydrated with different gradient concentrations of ethanol solutions (30%, 50%, 70%, 80%, 90%, 95%, 100%) for 15 min at each concentration. The sample was then treated with a mixture of isoamyl acetate and ethanol (1:1 *v*:*v*) for 30 min and treated with pure isoamyl acetate overnight. The processed samples were dehydrated by critical point method and coated with gold spray and photographed by a Hitachi Model SU8010 FASEM (Tokyo, Japan).

For transmission electron microscopy, the tissue was fixed, dehydrated, replaced with acetone, and infiltrated and embedded in Spurr resin (1:1 for 1 h and 1:3 for 3 h). Then, the samples were put into an oven to polymerize, solidify, and sliced by an ultramicrotome. After removing the slices with a 200-mesh copper mesh, they were double-stained with 2% uranyl acetate and 1% lead citrate. The processed samples were visualized via a Hitachi Model TEM 7800 (Tokyo, Japan).

### 2.8. Gas Chromatography−Mass Spectrometry (GC/MS) Analysis

Serum samples were prepared as follows: first, 50 μL of the serum sample was uniformly mixed with 10 μL of internal standard (L-2-chloro-phenylalanine, methanol), adding 150 μL of the methanol-acetonitrile (*v*:*v* = 2:1) solution and mixing. After ultrasonic extraction, the supernatant was placed at −20 °C for 10 min and centrifuged at low temperature for 10 min (12,000 r/min, 4 °C). The supernatant was taken into a derivative bottle, the samples were freeze-dried and then 80 μL the methoxyamine hydrochloride pyridine solution with a concentration of 15 mg·mL^−1^ was added, which was mixed, and the oxime reaction was performed at 37 °C for 1.5 h. Then, 80 μL of derived reagent (BSTFA containing 1% TMCS) and 20 μL of n-hexane were added, mixed, and reacted at 70 °C for 1 h, the samples were then determined by GC/MS. Then, 1 μL of the derivatized sample was injected into the Agilent 7890A-5975C GC/MS system (Agilent, USA), separated by the capillary column and entered into the mass spectrometry system for detection. The carrier gas used in this experiment was high-purity helium, and the flow rate was set to 1.0 mL·min^−1^. The temperature program was set as follows: the initial temperature was set to 60 °C, and the temperature was increased to 305 °C at a rate of 8 °C min^−1^, and maintained for 6 min. The electron impaction source (EI) temperature was set to 230 °C, the quadrupole temperature was set to 150 °C, and the electron energy was 70 EV. The scanning mode was full scan mode (SCAN), and the mass scanning range *m*/*z* was 50–600; continuous sample analysis was carried out in random order to minimize or avoid the influence caused by the fluctuation of the instrument signal.

### 2.9. Statistical Analysis

In this study, the original disembarking data of GC-MS were preprocessed by the ChromaTOF software, and then the matrix of data was exported, false positive peaks and internal standard peaks were removed. A three-dimensional data table including the mass-to-charge ratio, retention time and peak area was obtained. All peaks were corrected using QC samples, and the corrected data were imported into SIMCA-P 14.0 software package (Umetrics, Umea, Sweden) for multi-dimensional statistical analysis to screen variables. The principal component analysis (PCA) score diagram was used to directly represent the differences between groups, and orthogonal partial least-squares-discriminant analysis (OPLS-DA) showed significant changes in chemical markers before and after screening. Variables with variable important in projection (VIP) values greater than 1 were considered differential metabolites. A 200-response ranking test was used to evaluate model fit. OPLS-DA and *t*-tests were combined to screen the differential metabolites in the serum of snakeheads in the control group and the experimental group (VIP > 1, *p* < 0.05).

The other statistical analysis was performed using one-way ANOVA followed by the least significant difference (LSD) multiple-range test with SPSS Version 20.0 (SPSS Inc., Chicago, IL, USA). All data were expressed as the mean with SEM. A level of *p <* 0.05 was considered to be statistically significant unless indicated otherwise. The calculation formulae of the growth and feed utilization parameters are shown below:Survival rate (SR, %) = (number of final fish/number of initial fish) × 100
Specific growth rate (SGR, % day^−1^) = [(ln (final weight) − ln (initial weight)]/breeding day × 100
Weight gain rate (WGR, %) = (final weight − initial weight)/initial weight × 100
Condition factor (CF, g·cm^−3^) = weight/(length)^3^ × 100
Viscerosomatic index (VSI, %) = viscerosomatic weight (g)/whole body weight (g) × 100
Hepatosomatic index (HSI, %) = hepatic weight (g)/whole body weight (g) × 100
Intestosomatic index (ISI, %) = length of intestine/length of body × 100
Feed conversion ratio (FCR) = dry feed intake (g)/(final body weight−initial body weight) (g)
Feed intake (FI, %) = feed consumption (g)/[(initial weight + final weight)/2 × number of feeding days] × 100

## 3. Results

### 3.1. Growth Performance, Feed Intake, and Morphological Parameters

Growth performance, feed utilization and morphological parameters of snakeheads fed with different diets were shown in Table 2. The initial body weight (IBW) of all fish was not significantly different (*p* > 0.05) (69.70 ± 0.30 g), and the final body weight (FBW) was three times the initial weight after 60 days. FBW, WGR, SGR and FCR were significantly increased in snakeheads fed E2 and E3 diets compared to the control diet (*p* < 0.05). Neither SR nor morphological parameters showed any statistical differences among the dietary treatments (*p* > 0.05).

### 3.2. Serum Biochemical Parameters

The serum biochemical indicators of snakeheads fed with different diets are presented in Table 3. Snakeheads fed diet E2 had lower BUN, ALT, and TG than those fed with control diet (*p* < 0.05), while no differences occurred among snakehead in the control diet, E1 and E3 (*p* > 0.05). In addition, compared to the control diet, AST was significantly decreased by dietary MEC inclusion (*p* < 0.05). However, there were no statistically significant differences in GLU, TP, ALB and TC among all groups (*p* > 0.05).

### 3.3. Digestive Enzymes Activities and Related Genes Expression

The mRNA expressions of amylase, pepsinogen and trypsinogen were significantly upregulated (*p* < 0.05) in the MEC treatment groups (Figure 1A,C,E). The amylase activities of the hepatopancreases and stomachs of snakeheads fed with the E2 and E3 diets were higher than those fed with the control diet (*p* < 0.05), without statistically significant in intestine (Figure 1B). There was no significant difference in the pepsin activities between the hepatopancreas and stomach (Figure 1D). However, the E2 and E3 groups had significantly higher trypsin activities in the hepatopancreas and intestine than the control group (*p* < 0.05) (Figure 1F).

### 3.4. Glucose Metabolic Key Enzymes Activities and Relative mRNA Expression

Table 4 represents the glucose metabolic key enzymes activities. Compared to the control group, snakeheads fed with the MEC diets displayed significantly increased G6P and GCS activity (*p* < 0·05), and E2 groups enhanced FBP activity (*p* < 0.05), whereas the HK activity in the E2 and E3 groups was significantly improved (*p* < 0.05). However, no significant differences were found between the control, E1, E2 and E3 groups in other key enzymes, including PFK, PK, PC, PEPCK and G6PDH. Furthermore, the mRNA levels of HK, FBP, G6P and GCS in hepatopancreas were also upregulated in the MEC groups compared with the control group (*p* < 0.05), except for HK and GCS in the E3 group (Figure 2).

### 3.5. Hepatopancreas Morphology, and the Composition of Hepatopancreas and Dorsal Muscle

The histological changes of hepatopancreas tissue were assessed by H&E (Figure 3A) and oil red O (Figure 3B). Observations revealed that liver cells of snakeheads fed the MEC diets were normal, but the fat particles gradually increased with increasing MEC levels. As shown in Figure 3C, compared to the control diet, snakeheads fed with the E2 diet had significantly increased glycogen and ether extract (*p* < 0.05). The liver glycogen and hepatopancreatic fat contents in the MEC group were increased compared with the control group. Moreover, as for the composition of dorsal muscle, higher level of lipids in the MEC groups were found (*p* < 0.05), and no significant changes were observed in ash, protein and moisture (Figure 3D).

### 3.6. Gastrointestinal Morphology and Integrity

Digital images and statistical analysis of the H&E-stained sections of the gastrointestinal tract indicated that the villi of the E1 and E2 groups were more densely arranged and the overall shape was better, while the villi of the control and E3 groups were looser (Figure 4A,D,G,J). Additionally, the villous height of the caecus and foregut of snakeheads fed with the E2 diet was significantly higher than those fed with the control and E3 diets (*p* < 0.05) (Figure 5A,B). Moreover, the scanning electron micrographs of gastrointestinal showed that the microvilli in the control group were severely damaged and arranged irregularly, but those in the E1 and E2 groups were in good shape and arranged neatly and densely (Figure 4B,E,H,K). In addition, we further observed the microvilli of the gastrointestinal tract with transmission electron microscopy (Figure 4C,F,I)) and found that the microvilli morphology of the control group was worse than that of the E1 and E2 groups, which corresponded to the results of scanning electron microscopy.

### 3.7. GC-MS Statistical Analysis and Discriminate Metabolites

Appendix A shows the typical total ion chromatograms (TIC) detected by GC-MS of representative serum samples of snakeheads from the control and MEC treatment groups. In this experiment, the signal of the serum metabolite collected by the instrument was strong, and the peak retention time and capacity generated could meet the test requirements.

Figure 6 displays the results of PCA analysis of serum metabolites in the control and MEC groups. As can be seen from the figure, the valid samples belong to the 95% confidence interval range. PCA analysis (Figure 6A,D,G,J) showed that its interpretation rate was high, which confirmed the reliability of the analytical model. To obtain a better understanding of the role of the MEC on snakehead, based on the PCA model, the OPLS-DA model was established to further analyze the changes in serum metabolic patterns in the control and MEC groups. The OPLS-DA plots (Figure 6C,F,I,L) exhibited that the serum metabolites of snakehead in the control group and the MEC groups were well separated on the principal component coordinate axis, and differences did exist. Moreover, the robustness of the model was investigated by the method of 200 permutation tests. The results are shown in Figure 6B,E,H,K, where R2X = 0.306, R2Y = 0.991 and Q2 = 0.899, indicating that the model was stable and reliable.

A total of 164 metabolites were identified from the plasma of snakeheads. Differential metabolites in the control group versus the MEC groups were filtrated by OPLS-DA and Student’s *t*-test (selected VIP > 1 and *p* < 0.05). The differential metabolites were identified based on their retention time and accurate mass as well as available standard compounds. As a result, 15, 17, and 14 different metabolites were identified in the E1, E2, and E3 diets, respectively, compared with the control diet (Appendix A). Among all of the differential variables, there were seven common metabolites found among the MEC groups.

The levels of the amino acids proline (FC, 1.997, 1.853, 1.882), uric acid (FC, 1.883, 1.786, 1.864), glycine (FC, 1.869, 1.731, 1.807), taurine (FC, 1.881, 1.872, 1.884), and dithioerythritol (FC, 2.003, 1.996, 2.178) in the E1, E2, E3 groups were significantly higher than the control group, and the lower levels of 4-hydroxyphenylacetic acid (FC, 0.443, 0.392, 0.343) and sorbose (FC, 0.557, 0.217, 0.148) were observed in the E1, E2, E3 groups, respectively. In addition, aspartic acid and norleucine in the MEC groups also showed an increasing trend, but the difference was not significant. The pronouncedly changed metabolites were associated with glucose metabolism and amino acid metabolism.

## 4. Discussion

Exogenous enzyme preparations have been extensively used in the aquatic feed industry due to their low cost and high efficiency. In correlational studies, the potential of different exogenous enzymes as green feed additives have been discussed and their beneficial effects on growth performance and feed utilization of terrestrial and aquatic animals have also been confirmed [3,19,20]. In this study, the supplementation of the MEC increased the FBW, SGR, WGR and FCR of snakeheads compared with the control group. This was consistent with the previous findings by Ghomi et al. [21] and Hlophe-Ginindza et al. [22], who reported that supplementing the multienzyme complex consisting of xylanase and protease significantly improved the growth performance of tilapia (*Oreochromis mossambicus*) and great sturgeon fingerlings (*Huso huso*). Meanwhile, studies on tilapia (*Oreochromis niloticus × O. aureus*) also showed the growth-promoting effect of the MEC [23]. Moreover, in this study, the effect of the MEC on fish growth performance may be related to the improvement of nutrient absorption and digestion. This was well in line with Owsley et al. [24], who demonstrated that snakeheads fed with amylase and protease diet improved the absorption and utilization of feedstuff by enhancing the enzymatic hydrolysis capacity of carbohydrates and protein substances in the feed.

The metabolism of a fish and its health status are usually reflected by the blood biochemical indicators of the fish, which are closely related to the nutrient levels of the feed [25,26]. Liver function is mainly reflected by serum TG, ALT and AST parameters [27,28]. It is generally considered that the activity of TG, ALT and AST in the body is not high and maintains a relatively stable state. Only when the body has a disease or injury will the activity in the serum increase [29]. In this study, MEC supplementation, irrespective of its level, did not change plasma metabolite levels and transaminase activity, which is a sensitive indicator of health status, suggesting that MEC administration is safe for snakehead. The metabolites of nitrogenous substances in fishes were ultimately urea nitrogen [26]. The current study demonstrated that snakeheads fed with the MEC diet significantly reduced BUN content compared to control diets. This indicated that the MEC enhanced the utilization of nitrogenous substances by snakeheads, thereby promoting the synthesis of nutrients and increasing weight.

Several studies argued that the effects of exogenous enzymes and animal endogenous enzymes are synergistic, inhibitory and ineffective [30,31,32]. In this study, snakeheads fed with MEC diets demonstrated significantly improved activities of digestive enzymes via up-regulating the related genes expression (including amylase, pepsinogen and trypsinogen); we speculate that dietary supplementation with MEC improved the ability of the hepatopancreas to produce and secrete digestive enzymes, and increased the content of digestive enzymes in the digestive tract, thereby enhancing the enzyme activity—that is, exogenous enzymes and endogenous enzymes have a synergistic effect. Consistent with this hypothesis, previous reports have shown that dietary supplementation of enzymes capable of hydrolyzing starch and protein can further promote the body’s decomposition and absorption of nutrients after entering the digestive tract, providing a nutrient substrate for some endogenous enzymes, thereby promoting the secretion of endogenous enzymes [32,33]. Conversely, in sea bass, dietary exogenous enzyme supplementation did not increase digestive enzymes activities [30], whereas supplementation of the NSP enzyme complex in a hybrid tilapia diet had no effect on protease and lipases activities [31]. This discrepancy between the above studies may be related to differences in fish species and exogenous enzymes types.

The liver plays an important role in glucose metabolism, which in turn was used as a measure of liver function [34]. In this study, the activity and expression of HK, FBP and G6P were remarkably stimulated by the appropriate dietary MEC inclusion, suggesting that the liver can absorb glucose from the plasma, thereby enhancing the glycolysis and gluconeogenesis of snakeheads. This was supported by the fact that once glucose enters the blood, it was taken up by liver cells and converted into liver glycogen and then entered different metabolic pathways [35]. This confirms previously reported results in Atlantic salmon (*Salmo salar*) and gilthead seabream (*Sparus aurata*) juveniles, where feeding with a carbohydrate-rich diet increased the activity of HK, GK and FBR in the liver [36,37]. Concurrently, Jin et al. [38] found that glucose increased the mRNA expression of FBP and PEPCK involved in gluconeogenesis in the rainbow trout hepatopancreas. G6PDH is a key enzyme in the pentose phosphate pathway and is involved in the production of the NADPH required for fatty acid synthesis [35]. The results of this study exhibit that snakeheads fed the MEC diet improved the G6PDH activity, GCS activity and gene expression, while the liver glycogen content also displays a significant increase. Previous studies have suggested that high mRNA levels in hepatic GCS may indicate enhanced glycogen synthesis [39]. The MEC enhanced the liver glucose metabolism, which might be ascribed to the fact that the amylase in MEC promoted the absorption and digestion of carbohydrates in the feed, so that the snakehead can use more carbohydrates and enhancing the ability of the hepatopancreas to metabolize carbohydrates. Nevertheless, few studies have focused on the mechanism through which exogenous enzymes affect glycometabolism in aquaculture, which requires further investigation.

In addition, the liver is also the most important organ in nutrient metabolism. In order to maintain the normal physiological balance of fish, its integrity and function must be maintained [40]. In the present study, Oil Red O and H&E staining found that snakeheads fed with the MEC diet did not affect the morphological characteristics of the hepatopancreatic cells, and the liver cells were all normal, but the glycogen and fat content were significantly increased, consistent with the biochemical parameters. Compared with the control group, serum ALT and AST levels were decreased in the MEC groups, indicating that dietary supplementation with MEC could reduce liver damage and enhance glycometabolism and fat metabolism. Additionally, fish muscle is the largest tissue and is mainly used for human food [41]. The nutritional value of the muscle includes the deposition of protein and lipids in the muscle [42]. However, many previous studies revealed that the nutritional value of fish is affected by many factors, such as age, species, and environmental conditions [43,44]. The present results suggest that supplementation with appropriate MEC can increase the lipid content in snakehead muscles. Lipids are a valuable source of energy for fish [45]. Dietary supplementation of MEC could affect metabolic enzymes related to lipid metabolism, which is reflected in TG, TC, glucose metabolic, and relative mRNA expression.

There are few studies studying the effect of supplied enzyme diet on the digestive morphology. Improper dietary supplementation might cause the characteristic and morphological changes in the epithelial cells and reduced nutrient digestibility [46]. In this study, we found that the moderate multienzyme diets (E1 and E2 groups) could improve the gastric and intestinal morphology and integrity in snakeheads, while the high-level multienzyme diet (E3 group) showed a reverse effect. Our previous study indicated that carbohydrates could affect the structure of the digestive tract of the snakehead, while the high starch content in the feed reduced the villi height and increased the gap of the snakehead digestive tract [9]. The present results confirm that snakeheads fed with MEC, which include amylase, acid protease, and neutral protease, demonstrated an improved ability to digest and absorb feed and had a reduced level of gastrointestinal peristalsis and friction with feed, thereby reducing the residence time of feed in the gastrointestinal tract. Therefore, supplementation with enzyme preparations in animal diets is an effective strategy to protect their digestive tract, but at appropriate levels with proper inclusion.

In the present study, we explored the plasma metabolic profile of the snakeheads fed with a normal diet and MEC-supplemented diets using GC-MS techniques. Pronouncedly changed metabolites were mainly related to glucose metabolism and amino acid metabolism, which indicated that the MEC additives could improve the growth performance through the metabolism of the glucose and amino acid pathway. Amino acids are major nutrients in the diet and the basic units of proteins. Elevated levels of amino acids were observed in the serum of snakeheads in the MEC groups. In this study, we found three amino acids levels were approximately two times higher in MEC groups than control group, including proline, glycine and taurine. Additionally, other amino acids, such as aspartic acid and norleucine, also showed an increasing trend in MEC groups. Proline was regarded as an essential amino acid for fish growth [47,48]. Proline can be synthesized from arginine by arginase in all animals, but the rate of synthesis is very low in fishes [48]. Thus, adequate dietary proline is essential to maximize fish growth performance and feed efficiency [49]. Glycine has been found to participate in gluconeogenesis, fat digestion and other metabolisms [50], and also stimulates feed intake in many fishes [51]. Riley et al. [52] reported that glycine supplementation in the diet improved the efficiency of nutrient absorption and the growth performance of rainbow trout. Glycine and proline are highly abundant in collagen and elastin, which are important proteins in connective tissue and other tissues [53,54]. Therefore, the adequate supply of both glycine and proline is essential for maximum collagen synthesis, maximum growth performance, and the optimal health of fishes [48]. Taurine was added into the normal diet as a raw material, which plays critical roles in lipid metabolism, anti-oxidation and osmoregulatory responses and is not incorporated into proteins [50,55]. Gaylord et al. [56] showed that the addition of taurine to all-plant protein diets promoted growth and feed utilization efficiency in rainbow trout. As the gastrointestinal morphology and activities of digestive enzymes improved, higher levels of amino acids were observed, providing compelling evidence that MEC can improve the amino acid metabolism to promote fish growth.

## 5. Conclusions

In conclusion, the current study proved that the effective improvement of the MEC (especially in group E2) affected the growth performance of snakeheads, but excessive MEC (group E3) inhibited snakehead growth. Dietary supplementation with the MEC is of great benefit to improve the activities and expressions of digestive enzymes, and to protect the gastric and intestinal morphology. Due to the key enzyme activities of glycometabolism increasing, glycolysis, gluconeogenesis, and glycogen synthesis were enhanced, and higher hepatopancreas glycogen and lipid levels were found in the MEC groups. From the metabolic analysis, amino acid metabolism was the main pathway influenced by the MEC, with several amino acid levels being elevated.

## Figures and Tables

**Figure 1 animals-12-00380-f001:**
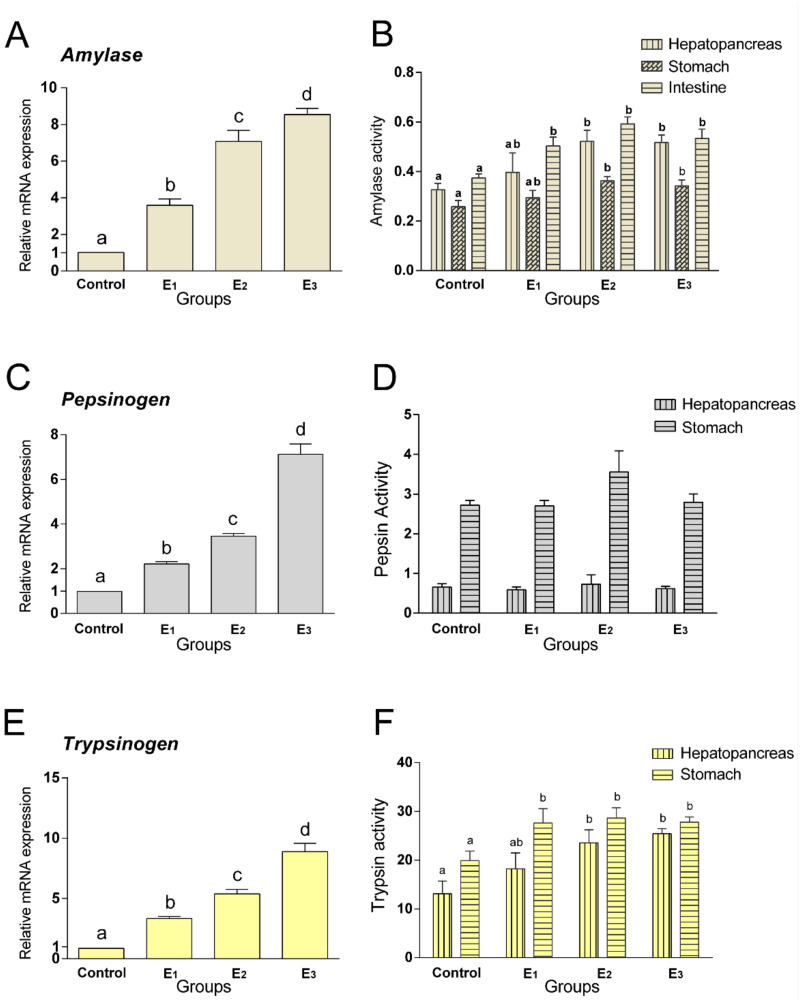
Relative mRNA expression (amylase (**A**), pepsinogen (**C**) and trypsinogen (**E**)) and the activities of digestive enzymes (amylase (**B**), pepsin (**D**) and trypsin (**F**)) of snakeheads fed experimental diets for 60 days. Data (mean ± SEM, *n* = 8) with different letters significantly differ (*p* < 0.05) among treatments.

**Figure 2 animals-12-00380-f002:**
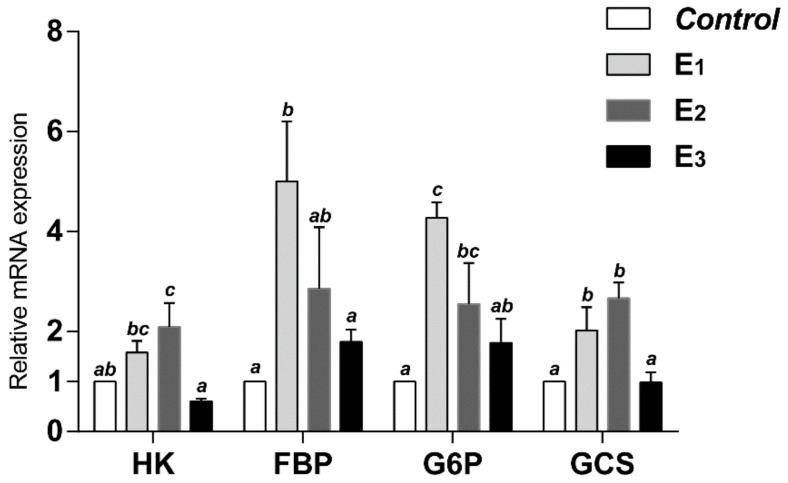
Relative mRNA expression of glucose metabolic key enzymes of snakehead fed experimental diets for 60 days. Data (mean ± SEM, *n* = 8) with different letters significantly differ (*p* < 0.05) among treatments.

**Figure 3 animals-12-00380-f003:**
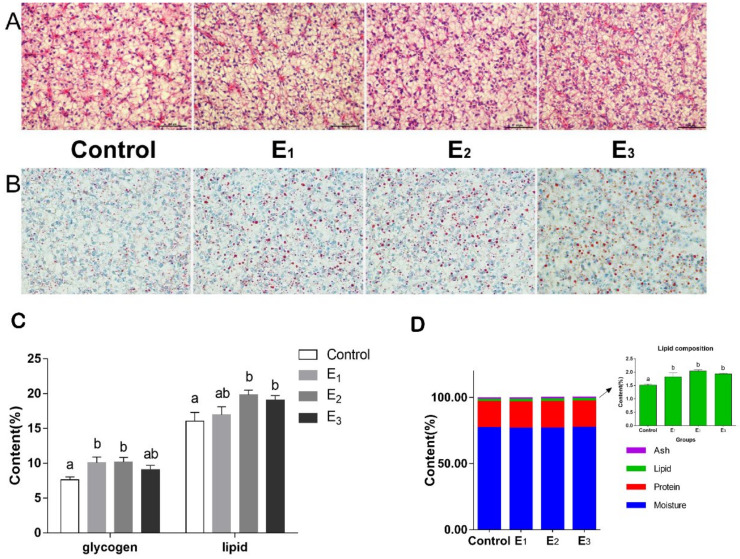
Hepatopancreas tissue stained by H&E (400×) (**A**) and oil red O (400×) (**B**), and the composition of the hepatopancreas (**C**) and dorsal muscle (**D**) of snakeheads. Data (mean ± SEM, *n* = 8) with different letters significantly differ (*p* < 0.05) among treatments.

**Figure 4 animals-12-00380-f004:**
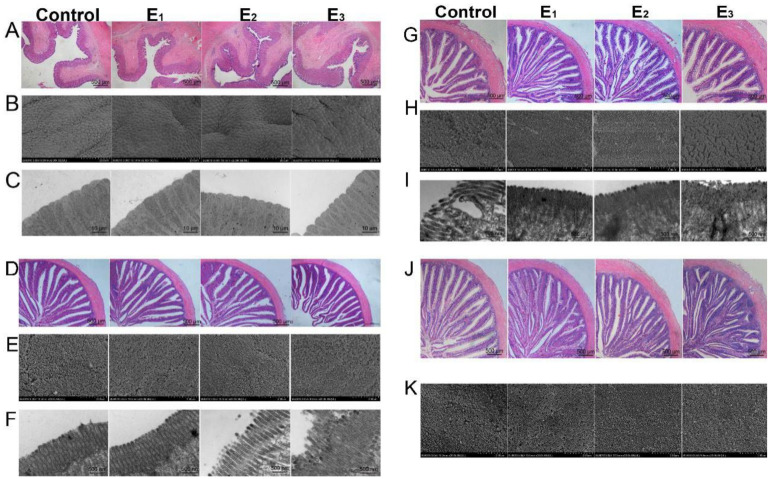
Gastrointestinal morphology of snakeheads fed experimental diets for 60 days. The stomach (**A**–**C**), caecus (**D**–**F**), foregut (**G**–**I**) and hindgut (**J**,**K**) were used to analyze intestinal morphology. (**A**,**D**,**G**,**J**) stained with H&E (40×); (**B**,**E**,**H**,**K**) were SEM images (2000×); (**C**,**F**,**I**) was TEM images (12000×). Data (mean ± SEM, *n* = 8) with different letters significantly differ (*p* < 0.05) among treatments.

**Figure 5 animals-12-00380-f005:**
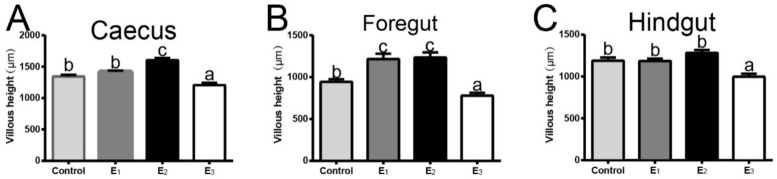
The villous height of the caecus (**A**), foregut (**B**) and hindgut (**C**) of snakehead fed experimental diets for 60 days. Data (mean ± SEM, *n* = 8) with different letters significantly differ (*p* < 0.05) among treatments.

**Figure 6 animals-12-00380-f006:**
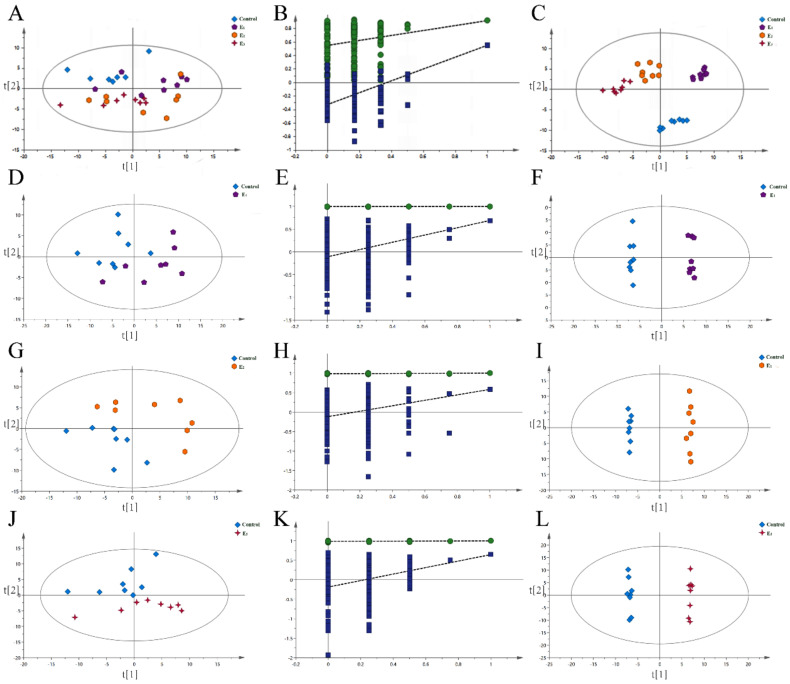
PCA score map (**A**,**D**,**G**,**J**), corresponding validation plots of OPLS−DA (**B**,**E**,**H**,**K**), and OPLS−DA score plots (**C**,**F**,**I**,**L**) derived from the GC/MS metabolite profiles for snakeheads fed Control and MEC-added diets. (**A**,**B**,**C**) indicate the totals. Data (mean ± SEM, *n* = 8) with different letters significantly differ (*p* < 0.05) among treatments.

**Table 1 animals-12-00380-t001:** Formulation and proximate composition of experimental diets (% in dry matter).

Ingredients %				Nutrient Levels ***	
Corn starch	5	Soybean oil	3	Crude protein	43.17
Wheat flour	15	Fish oil	2	Ether extract	13.5
Fish meal	40	Beer yeast	3	Crude fiber	1.76
Soybean meal	10.5	Bentonite	2	Ash	6.65
Peanut meal	9	Phosphatide oil	3	ME (MJ/kg)	2.49
Rapeseed meal	1	Ca(H_2_PO_4_)_2_	1.5	DE (MJ/kg)	2.13
SVO *	3	Premix **	2	GE (MJ/kg)	21.53
Total		100			

* SVO, squid visceral ointment. ** Premix, vitamin premix provided the following per kilogram of diets: VA, 8000 IU; VB12, 5 mg; VB6, 12.5 mg; VB2, 15 mg; VB1, 7.5 mg; niacin, 100 mg; choline chloride, 2500 mg; folic acid, 5 mg; inositol, 200 mg; biotin, 1 mg; D-pantothenic acid, 50 mg; VK3, 10 mg; VE, 300 mg; VD3, 2000 IU; VC, 240 mg; BHT, 400 mg; α-cellulose, 160 mg; MnSO_4_, 70 mg; ZnSO_4_, 100 mg; MgSO_4_, 300 mg; FeSO_4_, 600 mg; CuSO_4_, 10 mg; CoCl_2_, 0.5 mg Na_2_Se_2_O_3_, 5 mg; KI, 10 mg; L-carnitine, 1000 mg. *** Nutrient levels: GE, gross energy; DE, digestible energy; ME, metabolic energy.

**Table 2 animals-12-00380-t002:** Growth performance of snakeheads fed experimental diets for 60 days.

Items	Control	E_1_	E_2_	E_3_	SEM	*p* Value
IBW, g	69.77	69.83	69.49	69.97	0.149	0.755
FBW, g	220.28 ^a^	229.31 ^ab^	238.71 ^b^	236.34 ^b^	2.719	0.041
WGR, %	215.72 ^a^	228.53 ^ab^	243.51 ^b^	237.80 ^b^	4.054	0.006
FI, %	2.04	2.06	1.97	2.03	0.165	0.292
SGR, % day^−1^	1.92 ^a^	1.98 ^b^	2.06 ^c^	2.03 ^bc^	0.017	<0.01
CF, g cm^−3^	1.47	1.43	1.47	1.54	0.021	0.271
VSI, %	8.96	9.28	9.12	8.94	0.162	0.887
HSI, %	2.04	2.34	2.38	2.41	0.103	0.529
ISI, %	76.74	77.31	78.16	75.83	1.035	0.894
FCR	1.18	1.21	1.11	1.12	0.020	0.038
SR, %	100	97.33	98.67	97.33	0.513	0.202

Data (mean ± SEM, *n* = 8) with different letters significantly differ (*p* < 0.05) among treatments.

**Table 3 animals-12-00380-t003:** Serum biochemical parameters of snakehead fed experimental diets for 60 days.

Items	Control	E_1_	E_2_	E_3_	SEM	*p* Value
GLU (mmol/L)	18.56	17.41	16.01	15.59	0.579	0.260
TP (g/L)	37.28	37.42	40.87	42.34	1.032	0.217
ALB (g/L)	17.35	19.13	17.97	18.34	0.051	0.683
BUN (mmol/L)	16.48 ^a^	15.33 ^ab^	13.68 ^b^	14.63 ^ab^	0.712	0.025
ALT (U/L)	5.20 ^a^	4.52 ^ab^	3.45 ^b^	5.83 ^a^	0.544	0.044
AST (U/L)	8.90 ^a^	5.86 ^c^	4.70 ^c^	7.60 ^b^	0.781	0.048
TG (mmol/L)	2.28 ^a^	1.80 ^ab^	1.25 ^b^	1.82 ^ab^	0.178	0.020
TC (mmol/L)	6.05	5.59	5.10	6.03	0.254	0.521

Data (mean ± SEM, *n* = 8) with different letters significantly differ (*p* < 0.05) among treatments.

**Table 4 animals-12-00380-t004:** Activities of glucose metabolic key enzymes of snakeheads fed experimental diets for 60 days.

Items	Control	E_1_	E_2_	E_3_	SEM	*p* Value
HK, U/gprot	8.57 ^a^	10.93 ^ab^	13.24 ^b^	12.16 ^b^	0.620	0.026
PFK, U/mg	2.94	3.76	3.59	3.62	0.212	0.570
PK, U/gprot	49.47	42.96	42.82	46.13	2.713	0.395
PC, U/mgprot	36.09	25.10	35.16	29.29	2.289	0.299
PEPCK, U/mg	0.901	0.824	0.863	1.025	0.076	0.837
FBP, nmol/min/mgprot	0.900 ^b^	1.071 ^b^	1.848 ^a^	1.012 ^b^	0.138	0.038
G6P, nmol/min/mgprot	23.97 ^a^	37.35 ^b^	34.56 ^b^	33.21 ^b^	1.869	0.043
G6PDH, U/mgprot	1.551	1.852	1.735	1.392	0.094	0.352
GCS, U/mgprot	0.501 ^a^	0.824 ^b^	0.941 ^b^	0.830 ^b^	0.063	0.045

Data (mean ± SEM, *n* = 8) with different letters significantly differ (*p* < 0.05) among treatments.

## Data Availability

The datasets analyzed in the present study are available from the corresponding author on reasonable request.

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
