# Peer review of "Effects of Dietary Multienzyme Complex Supplementation on Growth Performance, Digestive Capacity, Histomorphology, Blood Metabolites and Hepatic Glycometabolism in Snakehead (Channa argus)"

_animals, 2022, doi:10.3390/ani12030380_

Round 1

Reviewer 1 Report

In this Manuscript, the authors aimed to explore the effects of multienzyme complex (MEC) inclusion with various levels in snakehead (Channa argus)) diets on growth performance, digestive enzyme activity, histomorphology, serum metabolism, and hepatopancreas glycometabolism. The subject itself is surely worthy of investigation. However, some points need to be addressed as follows:

  • Abstract: the recommended level of MEC had to be clearly stated at the end of the abstract. The total number of fish and their initial weights ± SE are missed.
  • According to the journal instructions, the abstract should consist of no more than two hundred words. Please minimize
  • The background about the importance of using each enzyme of MEC in aquaculture feeding is not sufficient.
  • The study's hypothesis should be clarified at the end of the Introduction section.
  • Lines 90-91: The MEC source (product, company, city, country) should be clearly stated.
  • Methods for estimating most of the nutrients in Table 1 were not found in the M&M section.
  • Line 111: Add more descriptions about experimental cages, such as their dimension. How much water was in each cage?
  • Lines 112-115: How did these doses come?
  • Have the water quality parameters have been evaluated?
  • Add the catalog number of the used kits for each parameter.
  • Line 217: LSD chosen here will increase the risk of type I error. Instead, the authors could use Duncan's multiple ranges or Tukey test.
  • In all Tables: describe the experimental groups used in the tables' footnotes.
  • The conclusion needs connections with the recommended level to take home massage for both scientific and practical.

Author Response

Dear Professors :

Thank you the reviewers for their constructive comments concerning our manuscript “Effects of dietary multienzyme complex supplementation on growth performance, digestive capacity, histomorphology, blood metabolites and hepatic glycometabolism in snakehead (Channa argus)” (ID: animals-1571282). Those comments are very valuable and helpful for revising our paper and guiding our researches. We have studied those comments carefully and have made corrections which we hope meet with approval. Revised portions are marked colored in the paper. The following is a point-to-point response to the editors' comments and recommendations. The main corrections in the paper and the response to the reviewer’s comments are as following:

Response to Reviewer 1 Comments and Reviewer 1’s comments were highlighted by using the Red Colored.

Point 1: Abstract: the recommended level of MEC had to be clearly stated at the end of the abstract. The total number of fish and their initial weights ± SE are missed.

Response 1: Thank you for your comment. We have clarified the optimal amount of MEC to add at the end of the abstract, and minimized the abstract word count according to the journal's requirements. In addition, the total number of fish and the initial weight have been added (Line 23-24, Line 37).

Point 2: The background about the importance of using each enzyme of MEC in aquaculture feeding is not sufficient.

Response 2: So far, there are not many applications of amylase, acid protease and neutral protease in aquafeed, and the complex of the three enzymes is even less, so there is no sufficient background information. However, related studies have reported the application of enzymes in livestock and poultry, such as Singh et al. showed that the combination of xylanase, amylase and protease can improve the utilization of nutrients. The study by Samuel et al. showed that feed supplementation with a multi-enzyme mixture can enhance the intestinal structure and immune performance of pigs.

Point 3: The study's hypothesis should be clarified at the end of the Introduction section.

Response 3: We have clarified the hypothesis at the end of the introduction section. Any change to our manuscript following your suggestion was highlighted by using the Red Colored Text (Line 70-72).

Point 4: Lines 90-91: The MEC source (product, company, city, country) should be clearly stated.

Response 4: The source of MEC has been supplemented (Line 84-85).

Point 5: Methods for estimating most of the nutrients in Table 1 were not found in the M&M section.

Response 5: Thanks for your suggestion, we have specified the compositional analysis of the experimental diet (Line 86-92).

Point 6: Line 111: Add more descriptions about experimental cages, such as their dimension. How much water was in each cage?

Response 6: Thanks for your suggestion, more descriptions about experimental cages have been added to the text (Line 109-111).

Point 7: How did these doses come?

Have the water quality parameters have been evaluated?

Add the catalog number of the used kits for each parameter.

Response 7: The optimal combination and dose of amylase, acid protease and neutral protease was designed by the Box-Behnken Design of Response Surface and determined by in vitro digestion. In addition, we have evaluated the water quality during the breeding experiment, mainly including (temperature, pH, dissolved oxygen value and transparency, etc.) (data not published). Besides, we have provided the names and manufacturer numbers of the chemical reagents used in the biochemical analyses (Line 130-135).

Point 8: Line 217: LSD chosen here will increase the risk of type I error. Instead, the authors could use Duncan's multiple ranges or Tukey test.

In all Tables: describe the experimental groups used in the tables' footnotes.

The conclusion needs connections with the recommended level to take home massage for both scientific and practical.

Response 8: In our current experiment, the control group has been defined, and what we need to do is a validation study, that is, the comparison between the experimental group and the control group. The sensitivity of the Least Significant Difference (LSD) is high, and the slight difference between the mean values ​​of each level may be detected. Therefore, the LSD multiple-range test is adopted. However, Duncan's Multiple ranges or Tukey test is more about pair-to-pair comparison between multiple averages, and the number of samples in each group is equal. Besides, we have made changes based on your suggestions and have combined the conclusions with the recommended doses (Line 495-496).

We tried our best to improve the manuscript and made some changes in the manuscript.  These changes will not influence the content and framework of the paper.

We appreciate for Editors/Reviewers’ warm work earnestly, and hope that the new version of the Manuscript fully matches to the requests of the Reviewers and of the Journal.

Once again, thank you very much for your comments and suggestions.

We look forward to hearing from you and to responding to any further questions and comments you and/ or the Reviewers may have.

Sincerely yours

Xaiqing Ding

on behalf of the authors

Reviewer 2 Report

The manuscript entitled "Effect of dietary multienzyme complex supplementation on growth performance, digestive capacity, histomorphology, blood metabolites and hepatic glycometabolism in snakehead (Channa argus)" signed by Xiaoqing Ding and colleagues is a simple and interesting work on the effect of multienzyme on fish diets.

The general structure of the work is simply and well described, and the text is clear and concise. Results are also interesting and promising.

In my opinion, some revisions are needed to make this article acceptable for publication:

  • Table 1: please specify if the Nutrient levels are theoretical or analyzed. Are those diet extruded or pelleted? Did you have analyzed the enzyme levels after the production? If yes, please specify the results.
  • 6; 164 metabolites were identified from the plasma, did you have found some metabolites with higher “weight” on the PCA? Normally the variables are also indicated in the PCA graph to see that some specific variables are predominant in the determination of the sample position in the plot. Some considerations were exposed in the discussion paragraph but not in the results.
  • Line 464, 465 and 466: feed not food.

Author Response

Dear Professors :

Thank you the reviewers for their constructive comments concerning our manuscript “Effects of dietary multienzyme complex supplementation on growth performance, digestive capacity, histomorphology, blood metabolites and hepatic glycometabolism in snakehead (Channa argus)” (ID: animals-1571282). Those comments are very valuable and helpful for revising our paper and guiding our researches. We have studied those comments carefully and have made corrections which we hope meet with approval. Revised portions are marked colored in the paper. The following is a point-to-point response to the editors' comments and recommendations. The main corrections in the paper and the response to the reviewer’s comments are as following:

Response to Reviewer 1 Comments and Reviewer 1’s comments were highlighted by using the Yellow Colored.

Point 1: Table 1: please specify if the Nutrient levels are theoretical or analyzed. Are those diet extruded or pelleted? Did you have analyzed the enzyme levels after the production? If yes, please specify the results.

Response 1: Thank you for your comment. The approximate composition of the diet was measured according to the AOAC (2005) standard method. All diets contained 49.5 g/kg moisture, 66.5 g/kg ash, 17.6 g/kg crude fibre, 135.0 g/kg ether extract, and 431.7 g/kg crude protein. Besides, the composition of the four diets and the total energy (21.53 MJ/kg) measured with an oxygen bomb calorimeter (1281, Parr, USA), the results of the compositional analysis of the experimental diets are shown as percentages on a dry matter basis on Table 1. The diets were processed into pellets of 5 mm in diameter with a steam puffing machine (DGP80-II; Yugong Technology Development Co., Ltd., Hebei, China). After pelleting, the relative additive amounts of the multienzyme complex were sprayed on the feed in accordance with the method of Wang et al. (2009). In addition, we analyzed the perzyme levels after production, and its amylase was 826.21 U kg-1, acid protease was 311.68 U kg-1, neutral protease was 3810.66 U kg-1 (Line 86-92, (Line 96-97).

Point 2: 164 metabolites were identified from the plasma, did you have found some metabolites with higher “weight” on the PCA? Normally the variables are also indicated in the PCA graph to see that some specific variables are predominant in the determination of the sample position in the plot. Some considerations were exposed in the discussion paragraph but not in the results.

Response 2: Thanks for your suggestion, there are 7 common differential metabolites in this study, namely sorbose, dithioerythritol, p-hydroxyphenylacetic acid, proline, glycine, taurine and uric acid, among which disulfide Erythritol, proline, glycine, taurine and uric acid are regulated to varying degrees, while 4-p-hydroxyphenylacetic acid and sorbose are down-regulated. And for the some considerations were exposed in the discussion paragraph but not in the results, because some metabolites such as aspartic acid and norleucine changed, but they did not reach a significant level, just a relative trend, we did not be specific in the results. Results have been refined based on your suggestions (Line 359-365).

Point 3: Line 464, 465 and 466: feed not food.

Response 3: According to your suggestion, we have changed food to feed (Line 461-463).

We tried our best to improve the manuscript and made some changes in the manuscript.  These changes will not influence the content and framework of the paper.

We appreciate for Editors/Reviewers’ warm work earnestly, and hope that the new version of the Manuscript fully matches to the requests of the Reviewers and of the Journal.

Once again, thank you very much for your comments and suggestions.

We look forward to hearing from you and to responding to any further questions and comments you and/ or the Reviewers may have.

Sincerely yours

Xaiqing Ding

on behalf of the authors

Reviewer 3 Report

The manuscript titled 'Effects of dietary multienzyme complex supplementation on growth performance, digestive capacity, histomorphology, blood metabolites and hepatic glycometabolism in snakehead (Channa argus)' is very well written and presents interesting results regarding the effects of supplementation on homeostasis of the study species.
Due to the interesting research question, I suggest that the authors make some corrections to improve the readability of the text: 

- Please add in the Introduction that Channa argus is an important invasive species
In Material and methods:
- Please indicate which type of amylase was measured (enzymatic and genetic analyses; in main text and in supplementary material)
- Please provide the names and manufacturer numbers of the chemical reagents used in the biochemical analyses.
- At what temperature were the biochemical analyses conducted?

- From which part of the intestine enzymatic analysis was performed?
- Lines 156-161 - Please add a description of the preparation of histological sections (embedding and cutting). Were the tissues embedded in paraffin?  Why was oil red staining performed? How were neutral lipids stabilized on histological sections to visualize them for oil red staining?

In Results:
Fig. 3B - why was oil red staining performed? What is the result? Please add this information to the text in the results. Please indicate in Fig. 3B what the red globules are
Figure 4 - please put bigger pictures. Details described in the text like microvilli are not well visible on miniatures.

Author Response

Dear Professors :

Thank you the reviewers for their constructive comments concerning our manuscript “Effects of dietary multienzyme complex supplementation on growth performance, digestive capacity, histomorphology, blood metabolites and hepatic glycometabolism in snakehead (Channa argus)” (ID: animals-1571282). Those comments are very valuable and helpful for revising our paper and guiding our researches. We have studied those comments carefully and have made corrections which we hope meet with approval. Revised portions are marked colored in the paper. The following is a point-to-point response to the editors' comments and recommendations. The main corrections in the paper and the response to the reviewer’s comments are as following:

Response to Reviewer 3 Comments and Reviewer ’s comments were highlighted by using the Blue Colored.

Point 1: - Please add in the Introduction that Channa argus is an important invasive species.

In Material and methods:

- Please indicate which type of amylase was measured (enzymatic and genetic analyses; in main text and in supplementary material)

- Please provide the names and manufacturer numbers of the chemical reagents used in the biochemical analyses.

- At what temperature were the biochemical analyses conducted?

Response 1: Thank you for your comment. We have added in the Introduction that Channa argus is an important invasive species (Line 42).

In this study, we measured α-amylase. We did not analyze the enzymology and genetics of this enzyme, but only determined the most active enzyme to be 826.21U/kg in the in vitro single factor digestion test by response surface methodology.

We have provided the names and manufacturer numbers of the chemical reagents used in the biochemical analyses. Besides, biochemical parameters were measured on ice at 4°C (Line 130-135).

Point 2: - From which part of the intestine enzymatic analysis was performed?

Response 2: Mid-hindgut of snakefish was used for the intestine enzymatic analysis.

Point 3: - Lines 156-161 - Please add a description of the preparation of histological sections (embedding and cutting). Were the tissues embedded in paraffin?  Why was oil red staining performed? How were neutral lipids stabilized on histological sections to visualize them for oil red staining?

Response 3: Based on your suggestion, we have added a description of the preparation of histological sections (embedding and cutting) to the text. First, the intestinal and hepatopancreas tissues were dehydrated by ethanol gradient method (75%, 85%, 95%, 100%), xylene twice for 10 min and paraffin wax three times for 60 min. The tissue in paraffin was embedded with an embedding machine. Then, the tis-sues sections of 4-μm thickness were sliced in a slicer. Flatten on glass slides and bake in a 60 °C oven. Hematoxylin staining for 15 min, eosin staining for 1 min, dehydration and mounting. Images were acquired using a DM3000 microscope (Leica, Wetzlar, Germany). The villus height was measured using Image-Pro software (Media Cybernetics, MD, USA).

We used oil red staining to detect the changes of fat particles, liver cells and fat content of snakehead.

The OCT embedded liver tissue samples were cut into frozen liver slices of 5 μm thickness. The slices were dried at room temperature for 2-3 min, then fixed in 10% cold formalin for 10 min. Washed 3 times with double distilled water. After that, transfer to 100% propylene glycol and place for 5 min (avoid bringing water into Oil Red O staining solution). Put the sections in preheated Oil Red O staining solution at 60 °C for 10 min. Counterstain with hematoxylin 2-3s, rinsed with tap water for 3 min, washed twice with double-distilled water, and stained sections were visualized and imaged with a DM3000 microscope (Leica, Wetzlar, Germany) (Line 157-169).

Point 4: Fig. 3B - why was oil red staining performed? What is the result? Please add this information to the text in the results. Please indicate in Fig. 3B what the red globules are

Figure 4 - please put bigger pictures. Details described in the text like microvilli are not well visible on miniatures.

Response 4: We used oil red staining to detect the changes of fat particles, liver cells and fat content of snakehead and added main information to the text in the results. Observations revealed that liver cells of snakehead fed the MEC diet showed normal, but the fat particles gradually increased with increasing MEC levels. Besides, The liver glycogen and hepatopancreatic fat contents in the MEC group were increased compared with the control group. The red globules in Figure 3B refer to the fat granules stained with Oil Red (Line 294-298).

Thank you very much for your suggestion. Figure 4 is composed of 44 single pictures. If the picture is too large, it will take up a lot of space. Therefore, for ease of reading, we have reduced the size of the images. We will provide the original images if the article is accepted for publication.

We tried our best to improve the manuscript and made some changes in the manuscript.  These changes will not influence the content and framework of the paper.

We appreciate for Editors/Reviewers’ warm work earnestly, and hope that the new version of the Manuscript fully matches to the requests of the Reviewers and of the Journal.

Once again, thank you very much for your comments and suggestions.

We look forward to hearing from you and to responding to any further questions and comments you and/ or the Reviewers may have.

Sincerely yours

Xaiqing Ding

on behalf of the authors

Round 2

Reviewer 1 Report

As the authors addressed the reviewer's comments, I suggest acceptance of the manuscript.